# Optical Fiber Vibration Signal Recognition Based on the Fusion of Multi–Scale Features

**DOI:** 10.3390/s22166012

**Published:** 2022-08-12

**Authors:** Xinrong Ma, Jiaqing Mo, Jiangwei Zhang, Jincheng Huang

**Affiliations:** Key Laboratory of Signal Detection and Processing, College of Information Science and Engineering, Xinjiang University, Urumqi 830017, China

**Keywords:** distributed optical fiber sensing, endpoint detection, feature fusion, differential pooling features, 2DCNN

## Abstract

Because of the problem of low recognition accuracy in the recognition of intrusion vibration events by the distributed Sagnac type optical fiber sensing system, this paper combines the traditional optical fiber vibration signal recognition idea and the characteristics of automatic feature extraction by a convolutional neural network (CNN) to construct a new endpoint detection algorithm and a method of fusing multiple–scale features CNN to recognize fiber vibration signals. Firstly, a new endpoint detection algorithm combining spectral centroid and energy spectral entropy product is used to detect the vibration part of the original signal, which is used to improve the detection effect of endpoint detection. Then, CNNs of different scales are used to extract the multi–level and multi–scale features of the signal. Aiming at the problem of information loss in the pooling process, a new method of combining differential pooling features is used. Finally, a multi–layer perceptron (MLP) is used to recognize the extracted features. Experiments show that the method has an average recognition accuracy rate of 98.75% for the four types of vibration signals. Compared with traditional EMD and VMD pattern recognition and 1D–CNN methods, the accuracy of the optical fiber vibration signal recognition is higher.

## 1. Introduction

With the rapid development of science and technology and the improvement in people’s living standards, people’s demands for a safer and more reliable perimeter security system have become more and more urgent. Optical fiber perimeter security is a technology with great development potential [1], which has become a hot field of international research. Optical fiber perimeter security systems have the advantages of long–distance and large–scale transmission, strong anti–electromagnetic interference ability, strong corrosion resistance, water resistance, and low cost. They have the irreplaceable advantages of other sensors and have been widely used in border monitoring, tunnel detection, offshore oil exploration, marine monitoring, seismic monitoring, oil pipeline monitoring and perimeter security, and other practical scenarios [2,3,4,5,6]. The sensing system based on Sagnac fiber optic interferometer has the advantages of small size, light weight, easy installation, and high sensitivity. There is no need for a reference fiber, and its symmetrical structure does not require high coherence of the light source, and the structure is simple, which is very suitable for distributed deployment of fiber vibration signal detection [7,8]. Therefore, in this paper, the recognition of the distributed Sagnac type optical fiber vibration signal is carried out.

There are three main steps in the recognition of optical fiber vibration signals: preprocessing of optical fiber vibration signals, feature extraction, and classification [9]. The preprocessing is mainly to denoise the vibration signal and detect the endpoints. Feature extraction is used to obtain different types of vibration signal features through different methods. At present, the effective features are the energy ratio, kurtosis, skewness, and spectral entropy of the signal, etc. Commonly used methods for feature extraction are short–time Fourier transform [10], empirical mode decomposition (EMD) [11] and variational mode decomposition (VMD) [12,13], and other methods. Then, the extracted optical fiber vibration signal features are classified and identified. Commonly used methods include naive Bayes discriminant function [14], support vector machine (SVM) [15,16,17], neural network [18,19], and other methods. In the naive Bayesian recognition method, the probability of each category is calculated according to the appearance of the feature attributes in each sample, and the highest probability is selected as the classification result of the sample. The main idea of SVM is to find an optimal decision hyperplane to maximize the distance between the two types of samples closest to the plane on both sides of the plane, so as to achieve the optimal classification of samples. The common method for neural networks to solve multi–classification problems is to set n output nodes, where n is the number of categories. The probability of any event occurring is between 0 and 1, and there is always an event (the sum of the probabilities is 1). The output of each node can be understood as the probability that the input signal is of this type, and the node with the highest probability is used as the classification result. Xu et al. used spectral subtraction to reduce broadband background noise and enhance the time–frequency characteristics of vibration signals. However, the spectral subtraction method is not ideal for denoising the low–frequency part of the vibration signal [20]. Tabi Fouda et al. used the short–term energy dynamic threshold method to discriminate and screen disturbance signals and classified and identified intrusion signals by extracting power distribution features [21]. Zhenshi Sun et al. carried out variational modal decomposition of the vibration signal, extracted the kurtosis statistical characteristics of each frequency mode, and combined with the zero–crossing rate characteristics of the vibration signal, used the SVM algorithm to analyze the time and frequency domains, the signal features are extracted in the time–frequency domain and other dimensions, and finally the five events are identified by the Hidden Markov Model [22]. However, the literature [21,22] relies on the manual extraction of signal features, and the manual extraction of features is uncertain, and whether the features of the original vibration signal can be fully extracted directly affects the classification accuracy of feature classification. Traditional optical fiber vibration signal recognition is mostly limited to the inherent mode of first extracting features and then classifying and identifying, which destroys the connection between the two and causes the loss of part of the signal information. In addition, the single–scale convolutional neural network can only extract part of the information of the signal and cannot extract the information of different scales of the signal.

In view of the above shortcomings in the recognition of optical fiber vibration signals, in this paper, we propose a new optical fiber vibration signal recognition method. A new endpoint detection algorithm combining spectral centroid and energy spectral entropy products effectively improves the endpoint detection effect, extracts features at different scales at different levels of the convolutional neural network, and a new method of differential pooling feature fusion to further improve feature accuracy, to better realize the classification and recognition of four different types of optical fiber vibration signals.

## 2. Distributed Sagnac Interferometer Fiber Sensing Technology

Angeline et al. [23] further developed Sagnac technology in the field of distributed optical fiber sensing based on fiber gyroscope (FOG) technology. The interferometer in the Sagnac interferometric fiber perimeter security system consists of a light source, a 3 dB coupler, a sensing fiber, and a light intensity detector [24]. The principle of distributed optical fiber sensing technology based on a Sagnac interferometer is shown in Figure 1.

The laser light emitted by the laser is divided into two parts by a 3 dB coupler, and the two separated beams propagate in the Sagnac fiber ring in two opposite directions, clockwise and counterclockwise, respectively, and interfere when the couplers meet. The split lasers start from the 3 dB coupler and arrive at the disturbance event point at different times. When they meet again, a phase difference will be generated at the 3 dB coupler. The external disturbance information can be obtained by demodulating the phase difference in the interference signal. If there is no corresponding disturbance information during the propagation process, the phase difference of the two separated laser beams remains unchanged, and a stable interference phenomenon occurs at the coupler. The Sagnac fiber optic vibration sensor does not require a reference fiber, and its symmetrical structure does not require high light source coherence, with simple structure, high reliability, and low cost in practical applications.

## 3. The Proposed Construction Method

The basic framework of optical fiber vibration signal recognition based on fusion of multi–scale features is shown in Figure 2. The original signal is preprocessed and intercepted, and the multi–scale feature extraction and classification recognition are integrated to realize the adaptive extraction and classification of optical fiber vibration signals. The model mainly includes signal preprocessing, multi–scale feature extraction, pooling structure, and classification recognition.

### 3.1. Data Preprocessing

The raw data of the optical fiber vibration signal contains many silent signals, which are useless. Therefore, in order to reduce the processing of silent signals and improve the operation rate, the end points of the original optical fiber vibration signals collected were detected and the vibration part of the signal was intercepted before the experimental data set was constructed. An overview of the optical fiber vibration signal preprocessing method is shown in Figure 3.

Signal endpoint detection is used to determine the starting point and ending point of the vibration part of the signal. The subsequent intrusion localization and event classification and recognition can be effectively realized only when the onset of vibration is accurately detected [25]. In a vibration signal containing environmental noise, the noise generally has a lower frequency spectrum and a lower spectral centroid, while the vibration signal has a higher frequency spectrum and a larger spectral centroid. Furthermore, the energy of the vibrating part of the signal is higher than that of the silent part. The spectral centroid algorithm and the short–term energy endpoint detection method can extract the signal vibration part from the raw data. However, an endpoint detection method combining spectral centroid and short–time energy has a good detection effect on vibration signals with large amplitudes, but not good on vibration signals with small amplitudes. The detection of spectral entropy vibration signal endpoints depends on the degree of disorder of the vibration signal but has nothing to do with its amplitude. Combining the short–term energy and spectral entropy to form the energy–spectral entropy product can improve the detection of weak vibration signals [26], but this method has certain misjudgment problems for weak signals. In addition, the traditional method uses a fixed double threshold for endpoint detection, which cannot completely detect the vibration signal.

In order to improve the accuracy of endpoint detection, the short–term energy and spectral entropy features are combined, and the spectral centroid method is introduced while forming the energy spectral entropy product. In this paper, a dynamic threshold endpoint detection method combining spectral centroid and energy spectral entropy product is proposed. This method not only overcomes the problem of unsatisfactory detection of small amplitude signals in the endpoint detection method combining spectral centroid and short–term energy but also overcomes the problem of misjudgment of weak signals by energy spectrum entropy products. Using dynamic thresholds can more fully extract the vibrating portion of the signal. The characteristic parameters of time domain and frequency domain are combined together, which can give full play to their respective advantages and effectively realize various vibration signals’ endpoint detection. The steps of the endpoint detection algorithm based on spectral centroid combined with short–term energy and spectral entropy are as follows:

(1)Framing the vibration signal. Since the signal after framing has the problem of energy leakage, in order to reduce the energy leakage, the framed part is windowed, and the Hamming window can be used to effectively reduce the energy leakage.(2)Calculate the spectral centroid of each frame of signal. The spectral centroid is the center of gravity of the spectrum. This feature is used to measure the position of the spectrum. Let the spectral centroid of the *i*–th frame be $C_{i}$, and the calculation formula of the obtained spectral centroid feature $C_{i}$ is:(1)$$C_{i}=\frac{\sum_{k=1}^{N}\left(k+1\right)X_{i}\left(k\right)}{\sum_{k=1}^{N}X_{i}\left(k\right)}$$
where $X_{i}\left(k\right)$ is the discrete Fourier transform of the *i*–th frame and *N* is the frame length.(3)Calculate the short–term energy of each frame of signal, set the short–term energy of the *i*–th frame as E(i), and the short–term energy calculation formula is:(2)$$E\left(i\right)=\frac{1}{N}{\sum_{n=1}^{N}\left|x_{i}\left(n\right)\right|}^{2}$$
where $x_{i}\left(n\right)$ is the signal of the *i*–th frame and *N* is the frame length.
Calculate the spectral entropy of each frame of signal, set the spectral entropy of the *i*–th frame to be $H_{i}$, and the spectral entropy calculation formula is:(3)$$H_{i}=\sum_{l=0}^{N/2}P_{i}\cdot \log P_{i}$$

Among them, $P_{i}$ and $H_{i}$ are the probability density and spectral entropy value of the *i*–th frame signal, respectively. Calculate the energy spectrum entropy product of the combination of short–term energy and spectral entropy of each frame of signal, set the energy spectrum entropy product of the *i*–th frame as $T_{i}$, and the calculation formula of the energy spectrum entropy product is:(4)$$T_{i}=\left(H_{i}-H_{0}\right)\cdot E\left(i\right)$$
where $H_{0}$ is the spectral entropy value of the background noise segment (0 in a noise–free environment); $H_{i}$ and E(i) are the spectral entropy and short–term energy of the *i*–th frame of the vibration signal, respectively.

(4)Median filtering is performed for the two feature sequences and a threshold is dynamically estimated, the histogram of each feature sequence is calculated and smoothed, and the local maximum value of the histogram is detected. Let $Y_{1}$ and $Y_{2}$ be the local maximum and sub–local maximum, respectively, and W is the parameter estimated by the threshold, then the calculation formula of the threshold T is:(5)$$T=\frac{W\cdot Y_{1}+Y2}{W+1}$$

(5)Use the thresholds of the two feature sequences to perform threshold judgment on each frame of signal. Suppose the spectral centroid threshold of the frame signal and the energy spectrum entropy product thresholds are, respectively, greater than the estimated feature sequence thresholds; the segment of the signal is considered to be a valid signal. According to the relationship between the frame and the frameshift, the position of the frame in the original signal is obtained. The comparison chart of different endpoint detection is shown in Figure 4; where (a) is the endpoint detection method of ordinary spectral entropy and short–term energy; and (b) is the endpoint detection method based on the combination of spectral centroid and energy spectral entropy product. Experiments show that the endpoint detection algorithm combining spectral centroid and energy spectral entropy product combines their respective advantages, reduces false alarms, and can effectively detect weak vibration signals to achieve accurate endpoint detection for various vibration signals.

The signal is intercepted according to the result of the threshold judgment, and it is judged whether the start position of the fiber vibration signal is detected. The sampling length is 80 k sampling points. Through empirical knowledge, we found that the data of 1 s time length can contain a complete vibration signal, and multiple short signals of running and walking can be regarded as a fiber–optic vibration signal for classification and recognition research. The interception of all data samples is repeated according to the above method. In order to facilitate subsequent data processing, the maximum value normalization is performed on all intercepted signal data. The four types of fiber vibration signal preprocessing results are shown in Figure 5.

### 3.2. Network Construction

In recent years, deep learning has achieved great success in image and speech recognition [27,28]. To more comprehensively display the comprehensive information of the original signal, improve the recognition rate, and reduce the error rate, the preprocessed optical fiber vibration signal is classified and recognized by a two–dimensional convolutional neural network. The purpose of the convolution operation is to extract features. Compared with other shallow or deep neural networks, CNNs need to consider relatively fewer parameters, and the hidden layers of CNNs perform input data in the filtered form. The 2DCNN network can automatically suppress noise in the signal and intelligently extract relevant features throughout the convolutional and pooling layers [29].

To more comprehensively display the comprehensive information of the original signal, improve the recognition rate, and reduce the error rate, a short–time Fourier transform, which maps one–dimensional time domain signals to two–dimensional features, is applied to the fiber vibration signal. The short–time Fourier transform has a certain degree of adaptability to the signal and can restore the energy distribution of the input signal in the time–frequency domain. Its calculation formula is:(6)$$STFT_{f}\left(t,\omega \right)=\int_{-\infty }^{\infty }f\left(t\right)g\left(t-\tau \right)e^{-j\omega t}dt$$
where f(t) is the time domain signal; g(t−τ) is the time window whose center is at time τ, and $e^{-jwt}$ is the modulation operator that converts the signal from the time domain to the frequency domain. In this paper, the Hamming window is selected to reduce the problem of spectral leakage. The window length determines the time resolution and frequency resolution of the spectrogram. The longer the window length, the longer the intercepted signal, the higher the frequency resolution after transformation, and the worse the time resolution. In order to overcome the correlation of the data, the adjacent windows partially overlap, the window length selected in this paper is 256, and the overlap is 128. In addition, it is converted into an RGB image of size 224 × 224 after short–time Fourier transform. Finally, the three–channel RGB image of the optical fiber vibration signal is used as the input of the two–dimensional convolutional neural network to realize the transformation from the spectrum perception problem of the one–dimensional optical fiber vibration signal to image recognition.

Influenced by the traditional optical fiber vibration signal identification method, combined with the ideas and methods proposed in [8,30], this paper constructs an optical fiber vibration signal identification network structure. The structure of the optical fiber vibration signal recognition network designed in this paper is shown in Figure 6. In order to improve the ability of the network to extract features, a three–channel convolutional neural network is used, and different scales of convolution kernels and pooling methods are used for the convolutional neural network of each channel to perform multi–scale feature extraction. The multi–scale features of the signal are fused to obtain more comprehensive feature information, which is helpful for the accurate classification and recognition of optical fiber vibration signals.

#### 3.2.1. Multiscale Feature Extraction

The application of a convolutional neural network to the recognition of optical fiber vibration signals can achieve accurate feature extraction and classification and recognition of different signals including vibration signals that are easily ignored due to weak energy. Usually, a single neural network cannot effectively extract all the information features of the vibration signal, and CNN extracts the features of the signal through a fixed convolution kernel size, and there is a problem of signal feature loss or misjudgment. In addition, pooling can reduce the size of feature maps and reduce the amount of computation, but there is also the problem of information loss. To this end, this paper uses convolutional neural networks of different scales to extract the features of vibration signals at multi–scale and multi–level. Different size convolution kernels and different pooling methods are used for the convolutional neural network of each channel, and the signal features of different hierarchical networks are fused.

Among the three convolutional neural networks of different scales, the convolutional neural network of the first channel adopts the ordinary convolutional neural network and the maximum pooling method for feature extraction; the convolutional neural network of the second channel adopts a parallel multi–branch spatial pyramid pooling structure to obtain features. The spatial pyramid pooling structure can solve the problem of repetitive feature extraction by convolutional neural networks, and extract features with different pooling kernel sizes [30]. The spatial pyramid pooling used in this paper is shown in Figure 7. It adopts a form of pooling equivalence and uses small–scale multiple pooling to equivalently obtain large–scale pooling, which improves computational efficiency. The convolutional neural network of the third channel adopts the differential pooling structure for feature extraction. In this network, the signal features of the second layer convolution, the third layer convolution, and the fourth layer convolution are extracted, respectively. The obtained signal feature information is fed into the differential pooling structure to obtain the corresponding differential pooling features. The convolutional neural network of the third channel adopts the differential pooling structure for feature extraction. The differential pooling structure can complement the features of different levels, and to a certain extent, the signal contains both semantic information and spatial information. Then, the multi–scale features extracted by the different scale convolutional neural networks of the three channels are converted into a unified dimension for fusion, and the fused features are sent to the MLP for classification and recognition, and the recognition result of the optical fiber vibration signal is obtained.

#### 3.2.2. Differential Pooling Structure

Maximum pooling can reduce the deviation of the estimated mean caused by the parameter error of the convolutional layer and retain more information. Average pooling can reduce the problem of increased variance of estimates caused by limited neighborhood size. However, the maximum pooling focuses more on extracting the local salient features of the vibration signal and the contour information of the vibration signal, and the pooled features cannot completely contain all the information of the vibration signal. For average pooling, although the pooled features can retain more background information, the calculation method is easily affected by background noise, and the distinction between vibration signals and background noise is not high. Therefore, a differential pooling structure is proposed, which combines maximum pooling and average pooling to make the extracted vibration signal features more comprehensive and discriminative, thereby improving the accuracy of optical fiber vibration signal recognition. The calculation formula of the differential pooling feature $P_{c}$ is as follows:(7)$$P_{c}=\theta_{1\times 1}\left(P_{avg}-P_{\max}\right)+\theta_{1\times 1}\left(P_{\max}\right)$$

Among them, $P_{avg}$ and $P_{\max}$ are the features after average pooling and maximum pooling, respectively; $\theta_{1\times 1}\left(x\right)$ is a 1 × 1 convolution kernel operation for the pooled feature x.

The differential pooling structure proposed in this paper is shown in Figure 8. The feature map after convolution undergoes average pooling and maximum pooling to obtain the pooled features, and the average pooling features of the extracted fiber vibration signals and the maximum pooling. The features are subtracted, and the subtracted features are convolved with a 1 × 1 convolution kernel to obtain the different features of the two. The extracted max–pooling features are convolved with a 1 × 1 convolution kernel and added to the different features obtained before to obtain the differential pooling features. The differential pooling structure combines the advantages of maximum pooling and average pooling and reduces the influence of background noise on the signal while obtaining the characteristic information of the fiber vibration signal.

## 4. Experiment and Analysis

The data used in this experiment came from the Sagnac-based interference-type perimeter security early warning system (Xinjiang Meite Intelligent Security Engineering Co., Ltd., Urumqi, China). The original data of the optical fiber vibration signal is collected by the Sagnac interferometric perimeter security early warning system combining the two layouts of hanging net and buried. Both the hanging net and the buried test environment are shown in Figure 9. Figure 9a is the hanging net test environment and Figure 9b is the buried test environment. The data collected in the experiment includes four types of optical fiber vibration signals: flapping, knocking, walking, and running. The sampling frequency of the acquisition system is 8 MHz. In the actual data acquisition, according to the preset parameters of the acquisition equipment, the equipment collects 80 k data points in 1 s, and the duration of each sampled data is about 3~4 s. After signal preprocessing, a total of 1569 signal samples were obtained, including 357 knocking samples, 338 flapping samples, 406 running samples, and 468 walking samples.

### 4.1. Network Training and Testing

In the experiments of this paper, the operating system is Windows 10 64 bit. The programming language is Python 3.8. The deep learning framework is PyTorch. We used the PyTorch platform to construct a fusion multi–scale feature network model, which was used to realize the construction of the network model. PyTorch is a fairly concise, efficient, and fast framework. Its advantages are that the design pursues the least encapsulation, the design is in line with human thinking, and it allows users to focus on realizing their ideas as much as possible. After many experimental parameter adjustments of the convolutional neural network model, the learning rate of the network was adjusted to 0.0001, the batch processing was 32, and the epoch was 100 times.

Through multiple experiments, parameters such as the dimension of the convolutional network, the number of convolutional layers, the size of the convolutional kernel, and the number of fully connected layer units were adjusted according to the experimental results. Different networks were compared and, finally, use of the fusion multi–scale feature convolutional neural network model was chosen. The details of the fusion multi–scale feature network are shown in Table 1. The input and output dimensions of the convolutional layer are expressed as: (number of feature maps)@(feature map size), for example (3@224 × 224).

In the convolutional network model, an RGB image with a size of 224 × 224 is input. First, three convolutional neural networks of different scales are used to extract the signal features of the input data. The convolutional neural network of each channel adopts different convolution kernels and pooling. The first CNN adopts a three–layer convolution and max–pooling structure; the second CNN adopts a three–layer convolution and spatial pyramid pooling structure for feature enhancement; the third CNN adopts a four–layer convolution and differential pooling structure, parallel multi–scale extraction of signal features. Batch normalization and ReLU activation function are used after convolution in the three–channel CNN. Second, the features extracted by the three–channel convolutional neural network are unified in dimension and feature fusion is performed. Finally, the fused features are output to two fully connected layers for classification, these two FC layers are equivalent to an MLP using hidden layers. The first fully connected layer contains 64 units, and a drop layer is added after the ReLU activation function to avoid data overfitting. The second fully connected layer implements the classification function and uses the SoftMax function to map the output of the second fully connected layer to a value between (0, 1), the sum of the four output nodes is 1, and the output of each node is understandable is the probability that the input signal is of this type, and the node with the highest probability is used as the classification result. In addition, the network uses the cross–entropy loss function to improve the generalization ability of the model and uses the Adam optimization algorithm to iteratively update the network to speed up the network convergence [31].

In this experiment, test samples are used to verify the classification accuracy of the fusion multi–scale feature network for four types of optical fiber vibration signals. The learning curve and confusion matrix of optical fiber vibration signal recognition results are shown in Figure 10. The learning curve shows how the accuracy and loss values for the training and validation datasets change over the course of training. It can be seen that with the increase in the number of iterations, the accuracy of the test set and the training set gradually increases, and the loss value gradually decreases. After about 58 iterations, the fusion multi–scale feature network begins to gradually converge, and the accuracy gradually stabilizes. The accuracy of the network training set is close to 100%, and the accuracy of the test set is stable at around 99%. The loss value for the training set drops to around 0.001, and the loss value for the test set drops to around 0.01. From the confusion matrix of the model, it can be seen that in 20% of the test samples, the recognition accuracy of the flapping samples is 99.02%, and 0.98% of the samples are mistaken for the knocking samples. The recognition accuracy of the knocking samples is 99.07%, and 0.93% of the samples are mistaken for the flapping samples. The correct recognition accuracy rate in walking samples was 99.18%, and only 0.82% of samples were mistaken for running samples. The recognition accuracy of running samples was 97.73%, and 2.27% were mistaken for walking samples. The average accuracy of the fused multi–scale feature model is 98.75%, and it can effectively distinguish four types of optical fiber vibration signals from flapping, knocking, walking, and running.

### 4.2. Experimental Comparison and Analysis

To verify the effectiveness of the method in this paper, comparative experiments were carried out using the collected data sets. The fused multi–scale features (Fusion–CNN) were compared with the first channel CNN (First–CNN), the second channel CNN (Second–CNN), and the dual–channel CNN (FS–CNN) in the experiments of this paper. The structures are shown in Table 2. For the fairness of the experiments, other classification methods use the same training and test sets as fused multi–scale features (Fusion–CNN), and the initial learning rate, epoch, and batch size are the same as those of fused multi–scale features (Fusion–CNN).

The performance indicators of different methods are compared in Table 3. Under the same conditions, the performance of fusion multi–scale features (Fusion–CNN) can be significantly improved, with an average recognition rate of 98.75%. Compared with the single–channel First–CNN method and the Second–CNN method, the performance indicators of the dual–channel FS–CNN method are significantly improved, which verifies the effectiveness of different scale CNN methods in extracting multi–scale features in the network. Compared with the FS–CNN method, the indicators of the fusion multi–scale feature (Fusion–CNN) method are improved, which shows that the differential pooling structure proposed in this paper combines the advantages of average pooling and maximum pooling. It is of great significance to improve the recognition performance of optical fiber vibration signals.

Furthermore, to test the model’s performance, fiber vibration signal identification methods, EMD decomposition, VMD decomposition, 1D–CNN, and fusion of multi–scale features, were compared. EMD decomposition and VMD decomposition extract features such as kurtosis, skewness, and energy ratio of the signal, combine multiple features, and use SVM for signal type classification. For the fairness of the experiments, the traditional SVM classification method uses the same training and test sets as fused multi–scale features (Fusion–CNN). Since SVM classification does not require many samples, the number of samples here is reduced by half. A total of 800 samples were used, and training and testing were randomly divided in a ratio of 8:2. Here, EMD and VMD pattern recognition refer to the signal after denoising and endpoint detection, and its classification accuracy is mainly affected by feature parameters. The 1D–CNN method uses the same training and testing sets as Fusion–CNN.

Table 4 shows the comparison experiments of the four optical fiber vibration signal classification methods, it contains the classification results for each sample type, the average accuracy of network classification, and the total number of parameters for each model. It can be seen that EMD decomposition and VMD decomposition have better recognition rates for flapping and tapping signals, but lower recognition rates for running signals. Not good for weak signal recognition. This also reflects that the traditional optical fiber vibration signal recognition method has the problem of information loss between feature extraction and classification recognition. Although the 1D–CNN method has a high average recognition rate, this method has a significant recognition error for the knocking signal and the flapping signal. It cannot accurately classify and recognize the optical fiber vibration signal. Compared with other methods, the fusion multi–scale feature (Fusion–CNN) method has better feature extraction ability and higher recognition effect, which effectively improves the robustness and recognition accuracy of the network.

## 5. Conclusions

In this paper, a new method for fiber vibration signal recognition based on endpoint detection and fusion of multi–scale features was proposed. The endpoint detection algorithm based on the combination of spectral centroid and energy spectral entropy product combines their respective advantages, which can have a good detection effect on low–frequency signals and low–frequency parts of signals and improve the detection accuracy of optical fiber vibration signals. The signal intercepted for 1 s can effectively distinguish the walking signal and the running signal which are difficult to distinguish due to the short vibration time. The method of fusing multi–scale features is used to extract the multi–scale and multi–level feature information of optical fiber vibration signals more comprehensively, and MLP is used for classification and recognition. In the experimental part, through the analysis of the network learning curve and confusion matrix, it is found that its average accuracy on the test set can reach 98.75%. The effectiveness of the fusion multi–scale feature method is proved by experimental comparison. Through the comparison of EMD and VMD traditional pattern recognition methods, 1D–CNN and fusion multi–scale feature methods, it is proved that the average accuracy of the method proposed in this paper is higher than the other three methods, which can greatly reduce the false alarm rate of optical fiber vibration signal recognition, and effectively identify four types of optical fiber vibration signals for flapping, knocking, walking, and running.

## Figures and Tables

**Figure 1 sensors-22-06012-f001:**
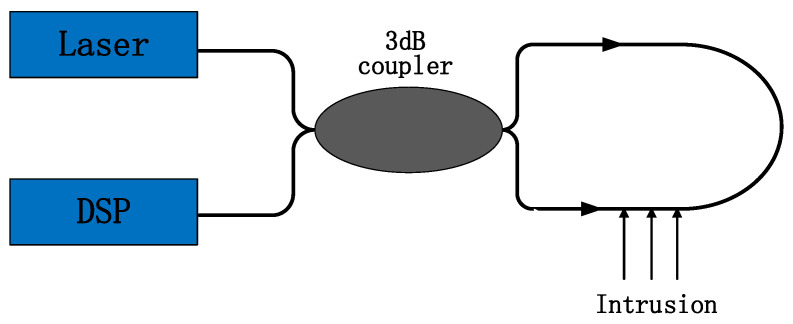
Schematic diagram of distributed optical fiber sensing technology based on a Sagnac interferometer.

**Figure 2 sensors-22-06012-f002:**
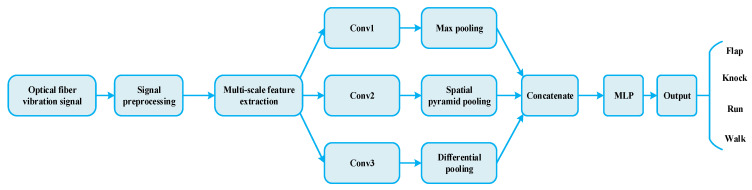
The overall structure of optical fiber vibration signal recognition based on fusion of multi–scale features.

**Figure 3 sensors-22-06012-f003:**
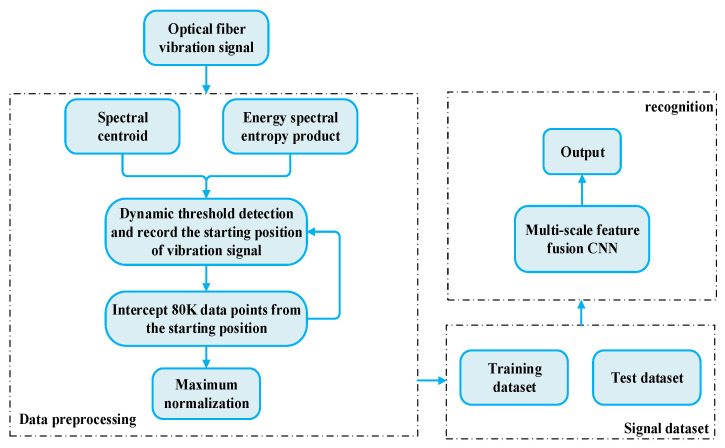
Overview of optical fiber vibration signal preprocessing method.

**Figure 4 sensors-22-06012-f004:**
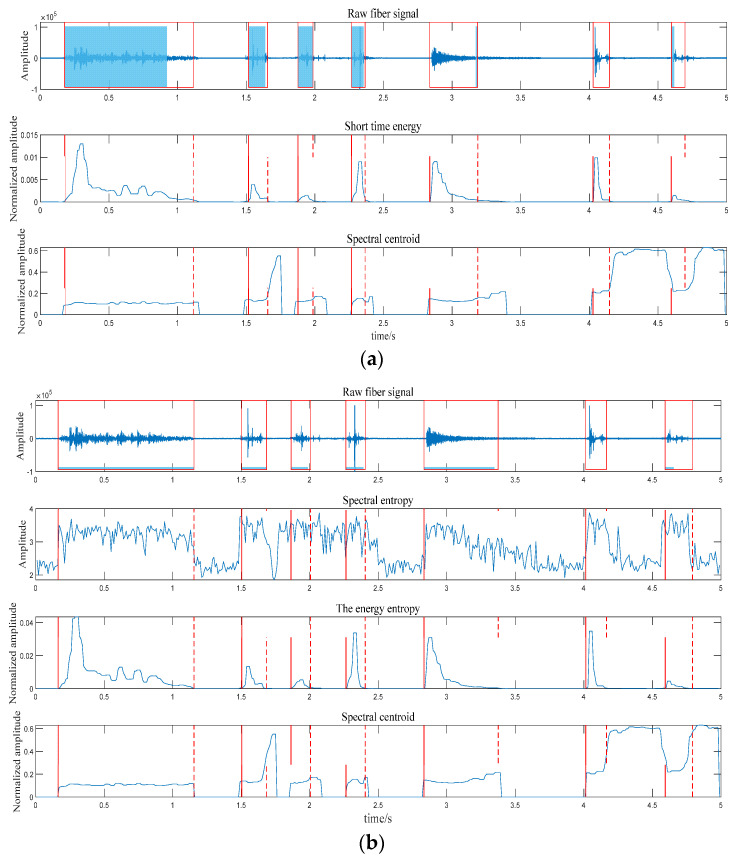
The comparison chart of different endpoint detection. (**a**) The endpoint detection method of ordinary spectral entropy and short–term energy; (**b**) the endpoint detection method based on the combination of spectral centroid and energy spectral entropy product.

**Figure 5 sensors-22-06012-f005:**
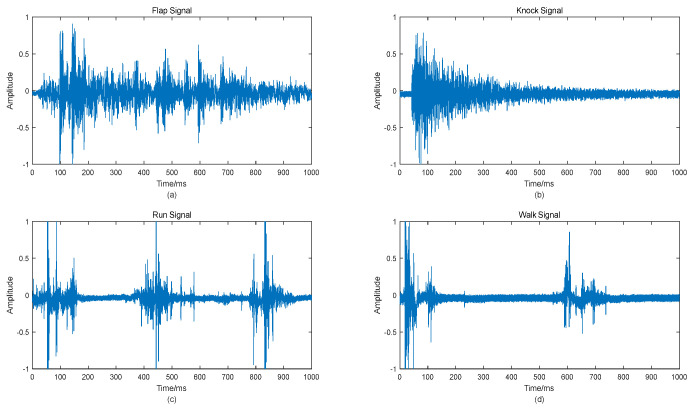
Four kinds of fiber vibration signals after preprocessing: (**a**) Flap signal; (**b**) knock signal; (**c**) run signal; (**d**) walk signal.

**Figure 6 sensors-22-06012-f006:**
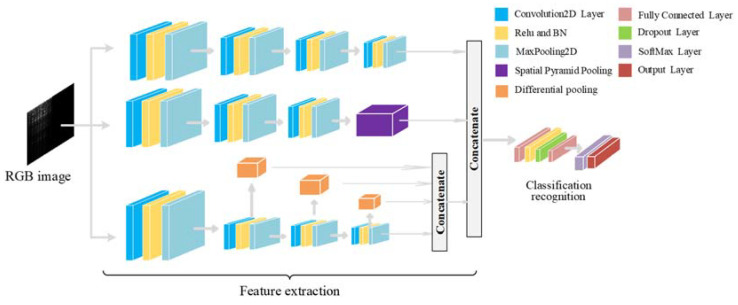
The structure of the optical fiber vibration signal recognition network.

**Figure 7 sensors-22-06012-f007:**
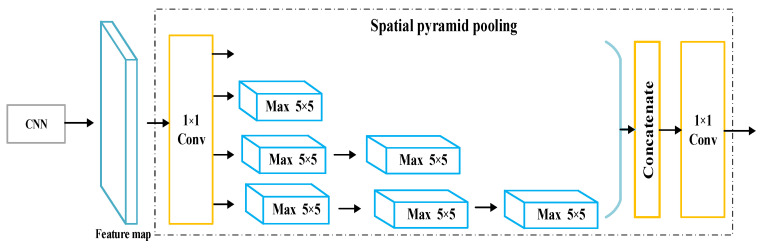
Spatial pyramid pooling structure.

**Figure 8 sensors-22-06012-f008:**
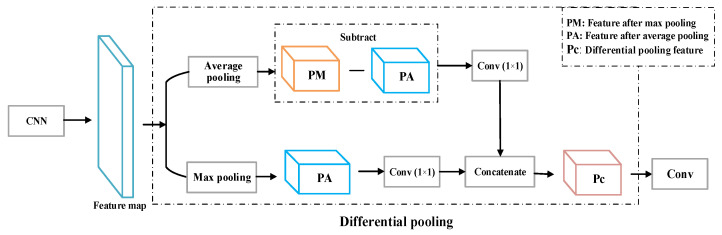
Differential pooling structure.

**Figure 9 sensors-22-06012-f009:**
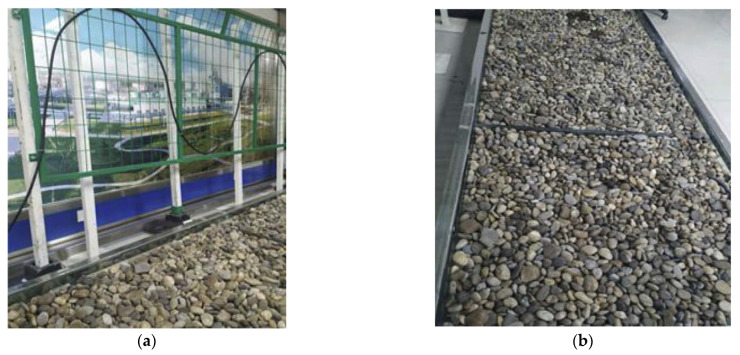
Both the hanging net and the buried test environment. (**a**) The hanging net test environment; (**b**) the buried test environment.

**Figure 10 sensors-22-06012-f010:**
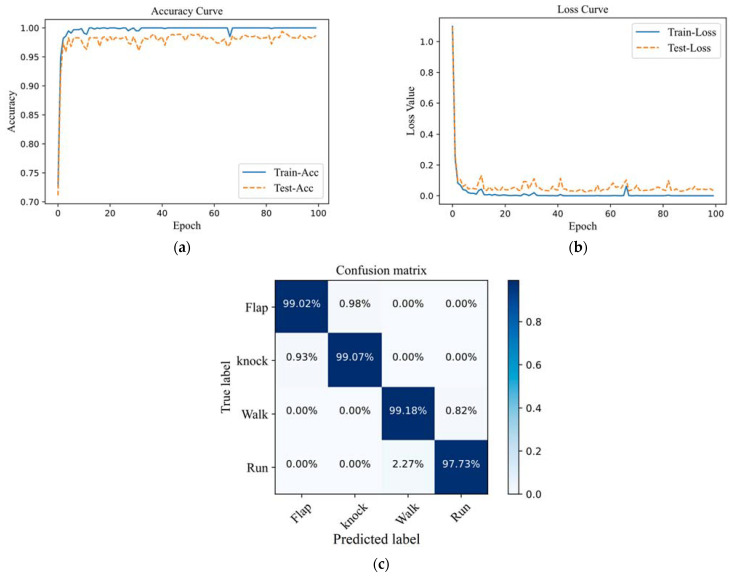
Learning curve and confusion matrix of fiber vibration signal recognition results. (**a**) Accuracy curve; (**b**) loss curve; (**c**) confusion matrix.

**Table 1 sensors-22-06012-t001:** Details of fused multi–scale feature convolutional neural network.

	First Branch		Second Branch		Third Branch	
Input	-	3@224 × 224	-	3@224 × 224	-	3@224 × 224
C1	16@5 × 5	16@220 × 220	16@3 × 3	16@222 × 222	16@1 × 1	16@224 × 224
P1	2 × 2	16@110 × 110	2 × 2	16@111 × 111	2 × 2	16@112 × 112
C2	128@3 × 3	128@54 × 54	128@3 × 3	128@55 × 55	128@3 × 3	128@111 × 111
P2	2 × 2	128@27 × 27	2 × 2	128@27 × 27	-	-
C3	64@3 × 3	64@13 × 13	64@3 × 3	64@13 × 13		64@55 × 55
P3	-	-	Spatial pooling		-	-
C4	-	-	16@13 × 13	16@13 × 13	-	128@55 × 55
-	-	-	16@13 × 13	16@13 × 13	-	-
	-	-	64@13 × 13	16@13 × 13	-	-
Concatenation						
Dense layer				64 neurons + dropout		
Output				4 neurons		

**Table 2 sensors-22-06012-t002:** Description of the different model structures.

NO.	Network	Description	Main Features
1	First–CNN	2D Standard Convolutional Neural Network	Network with standard convolutional kernel and maximum pooling and a fully connected layer in the last layer.
2	Second–CNN	2D–CNN–Spatial pooling–FC	The first layer of the network uses standard convolution kernels and max pooling, and the last layer uses spatial pyramid pooling.
3	FS–CNN	Parallel–2D–CNN–FC	The first channel CNN adopts normal network and max pooling, and the second channel CNN adopts spatial pyramid pooling.

**Table 3 sensors-22-06012-t003:** Comparison of performance indicators of different methods.

Method Classification	First–CNN	Second–CNN	FS–CNN	Fusion–CNN
Knock signal	71.43%	75.0%	89.08%	99.07%
Flap signal	98.40%	98.36%	98.15%	99.02%
Walk signal	100%	98.33%	96.43%	99.18%
Run signal	78.46%	80.0%	97.54%	97.73%
Average accuracy	87.07%	87.92%	95.30%	98.75%
Parameters	1.49 M	1.56 M	1.92 M	2.68 M

**Table 4 sensors-22-06012-t004:** Comparison of experimental results of three classification methods.

Method Classification	EMD	VMD	1D–CNN	Fusion–CNN
Knock signal	90.0%	99.33%	93.44%	99.07%
Flap signal	100%	100%	94.64%	99.02%
Walk signal	71.62%	95.95%	100%	99.18%
Run signal	56.14%	73.60%	98.46%	97.73%
Average accuracy	79.44%	92.22%	96.64%	98.75%

## Data Availability

The data presented in this study are available on request from the corresponding author.

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
