# Peer review of "Optical Fiber Vibration Signal Recognition Based on the Fusion of Multi–Scale Features"

_sensors, 2022, doi:10.3390/s22166012_

Round 1
Reviewer 1 Report
In this paper, the authors combine the traditional optical fiber vibration signal recognition idea and the characteristics of automatic feature extraction by a convolutional neural network (CNN) to construct an endpoint detection algorithm and a method of fusing multiple-scale features CNN to recognize fiber vibration signals. Experiments indicate that the method present here has an average recognition accuracy rate of 98.75% for the four types of vibration signals. This article is clear, concise, and suitable for the scope of the journal. Several small suggestions are supplied:
1. Suggest the authors supply more detail about the experiment setup.
2. Suggest the authors supply more detail about the fiber vibration signal recognition results.
3. The sensor present here may also be useful for marine monitoring, suggest the authors enhance the introduction part with one lastest review on this topic:
Optical fiber sensing for marine environment and marine structural health monitoring: A review Optics and Laser Technology, 2021.
Author Response
Thanks a lot for your comments and suggestions. Please refer to the attachment.

Reviewer 2 Report
The paper subject is determination of vibration signals based on CNN in distributed optical fiber sensing method.
Remarks:
1. Page 2. “Feature extraction is to obtain different types of vibration signal features through different methods”. What types of vibration signal features were determined using listed methods?
2. Page 2. How the vibration signal features were classified and identified using listed methods?
3. Fig. 3. The scheme is unclear. The raw data contains both noise and the vibration influence effects. If I understand correctly, the aim of the signal preprocessing is to extract vibration part or remove the additional features from the signal. The unclearness is an effect of mixing methods (e.g. short-time energy) and products (effects) of the methods (energy spectral entropy product) and signals (training dataset). Due to this, the scheme part for Endpoint detection suggests that something will be added to the signal and it is hidden under: spectral centroid, short-time energy and the spectral entropy. Please reconsider the scheme.
4. Page 4. Please highlight what is exactly new in the proposed method as the listed procedures are known.
5. Fig. 4. The raw signals in a an b cases seems to be slightly different.
6. Section 3.2. A more detailed description of the CNN is required. How is the signal prepared before using neural network? Why the signal in the form of picture is used? How the parameters of the layer are determined? Why such architecture was used?
Minor remarks:
1. page 2, line 50. et al. is before and after the name Xu.
2. Page 7.“In different convolutional neural networks, different layers of convolutional layers will generate different feature maps, and the information contained in the feature maps of each layer is also different.” It is true, but what kind of information is presented here to the reader?
Author Response

(The authors gave the same response as above.)
